# Routine Immunisation Coverage Shows Signs of Recovery at Global Level Postpandemic, but Important Declines Persist in About 20% of Countries

**DOI:** 10.3390/vaccines13040388

**Published:** 2025-04-03

**Authors:** Beth Evans, Laurent Kaiser, Olivia Keiser, Thibaut Jombart

**Affiliations:** 1Institute of Global Health, Faculty of Medicine, University of Geneva, 1202 Geneva, Switzerland; 2Division of Infectious Diseases, Geneva University Hospitals, 1205 Geneva, Switzerland; 3MRC Centre for Global Infectious Disease Analysis, School of Public Health, Imperial College London, London SW7 2AZ, UK; thibautjombart@gmail.com

**Keywords:** routine immunisation, COVID-19, pandemic, disruption, global

## Abstract

**Background/Objectives:** Routine immunisation (RI) coverage declines during the COVID-19 pandemic, from 2020 to 2022, are well-reported. With the declared end to the Public Health Emergency of International Concern in May 2023, and the cessation of most nonpharmaceutical interventions that were introduced to prevent or minimise COVID-19 spread, we (I) assess whether routine immunisation coverage has rebounded to the level of prepandemic trends and (II) seek to identify factors that help predict whether country performance has exceeded, maintained, or declined compared with expectations (based on time-series forecasting). **Methods:** We quantified global and country-level routine immunisation diphtheria–tetanus–pertussis (DTP) coverage trends postpandemic (2023) compared with prepandemic trends using time-series forecasting across 190 countries. We used discriminant analysis of principal components and random forests to identify relevant predictors of country-level coverage performance, including twenty-eight indicators of health system strength, health workforce, country income, pandemic containment, economic and health policies, and demographic aspects. **Results:** We show that mean global DTP third-dose coverage levels remained on average 2.7% [95% confidence intervals: 1.1–4.3%] lower than expected in 2023. However, once accounting for temporal demographic changes, we find that this translated to the total number of immunised children almost reverting to expected levels because of decreasing fertility reducing global-level immunisation target populations. At the country level, notable disruption remained in over thirty countries (16.8% of countries below expectations, 81.6% within expected ranges, and 1.6% above expectations). Neither predictive method performed well at identifying factors associated with coverage disruptions. **Conclusions:** Despite the end of COVID-19 pandemic measures, RI remains below expectations in about 20% of countries. No clear drivers of this continued disruption were identified. Further research is required to inform recovery efforts and prevent future epidemic and pandemic disruptions to routine health services.

## 1. Introduction

Routine immunisation (RI)—vaccinations given to infants typically aged 24 h to 18 months old—is an essential preventative health intervention, estimated by the World Health Organisation (WHO) to prevent 3.5 to 5 million deaths per year [1]. Declines or disruptions to RI coverage risk vaccine-preventable deaths and disease outbreaks [2]. Such disruptions and declines occurred during the SARS-CoV-2 (COVID-19) pandemic from 2020 to 2022, according to both health worker and parental/guardian surveys [3,4,5] and quantitative modelling [6,7]. Global trends indicated coverage declines in 2020, reaching an almost twenty-year low in 2021, followed by tentative hints of recovery in 2022 [7]. Understanding if and where disruptions remain is important to helping target finite resources for catch-up to mitigate risk of vaccine-preventable deaths or outbreaks.

When considering immunity gaps from reduced coverage, understanding the number of missed immunisations is important to help inform catch-up activities, e.g., quantifying additional vaccine requirements and/or targeting efforts in areas with the most missed children. WHO and UNICEF Estimates of Immunisation Coverage (WUENIC) report coverage annually in terms of percentage coverage [8,9] rather than absolute numbers of immunisations. Thus, changes in demography—including dramatic declines in global fertility rates [10]—must be combined with coverage levels to assess immunisation performance in terms of missed populations. Here, we use WUENIC coverage and United Nations World Population Prospects (UNWPP) data [11] to account for demographic changes when exploring country-level and global-level RI performance postpandemic.

Understanding the drivers of immunisation disruption is crucial to implementing effective countermeasures and preventing such declines in the future. During the COVID-19 pandemic, survey respondents cited increased vaccine hesitancy, reduced capacity due to focussing limited immunisation resources on COVID-19 vaccination campaign efforts, health system closures, increased difficulty travelling to health facilities, fear of catching COVID-19, and/or lockdowns as drivers of disruption [12,13]. These factors typically ended by 2023: the WHO declared an end to the COVID-19 Public Health Emergency of International Concern (PHEIC) on 5 May 2023 [14], and most countries ended or had already reduced pandemic response policies and COVID-19 vaccine mandates. However, it remains unclear whether the easing or ending of these restrictions and requirements enabled RI coverage to revert to prepandemic trends in the first year after the pandemic.

Here we aimed to:(1)Quantify global and country-level immunisation coverage trends in the pandemic and postpandemic period (years: 2020–2023) compared with prepandemic (years: 2000–2019) in terms of percentage coverage and number of immunisations;(2)Explore potential predictors that help in understanding routine immunisation country performance compared with expectations, factoring in country demographics, RI programme breadth, health financing, pandemic health system disruption, and COVID-19 policy responses.

## 2. Materials and Methods

### 2.1. Coverage Trends

We use our previously published methodology for forecasting expected coverage in the absence of disruptions in 2020–2023, updated for the most recent WUENIC data (published July 2024, [15]). In brief, we used autoregressive integrated moving average (ARIMA) time series forecasting [16] based on coverage trends since 2000 to project 2020–2023 coverage levels per country for all 190 countries in the WUENIC dataset with complete time-series data from 2000–2019 inclusive. We then calculated coverage deltas (defined as reported minus modelled coverage, i.e., a negative coverage delta indicated coverage lower than expected) for this period. We did this for diphtheria–tetanus–pertussis (DTP) first-dose (DTP1), to understand trends in reaching zero dose children—those who receive no vaccinations [17] and are considered essential for reaching Sustainable Development Goals, and third-dose (DTP3), to understand broad immunisation system performance trends, in line with WHO guidance [18].

### 2.2. Global and Country-Level Performance

To assess global coverage trends, we (1) conducted *t*-tests on coverage deltas per year across the same 190 countries as in Section 2.1 and (2) translated coverage trends into number of immunisations per year (based on UNWPP surviving infant estimates [11]) and calculated the corresponding *t*-tests on deltas in number of immunisations. For both, the hypothesis (H_0_) was that the delta is zero. To assess country-level performance, countries were classified into three categories based on 2023 performance: either below expectations (reported coverage < expected) and outside 95% ARIMA confidence intervals, within expectations, or above expectations (reported coverage > expected) and outside 95% ARIMA confidence intervals.

### 2.3. Predicting RI Performance

We consolidated twenty-eight descriptors from a range of publicly available nationwide datasets aiming to describe (i) variations in country health systems and service delivery and (ii) major pandemic factors that may have disrupted health service delivery—see Table 1. The health system and financial variables were aimed at quantitatively summarising prepandemic immunisation and health system quality, performance, financial investment, and pandemic preparedness. The pandemic impact variables were aimed at synthesising potential key disruptors to routine immunisation delivery. Population size was included in case differently sized countries experience disruption from different combinations of drivers.

Data cleaning involved summarising prepandemic indicators as the mean of 2015 to 2019 values (unless not available, in which case 2019 data were used). For indicators reporting variables during the pandemic period, we synthesised more granular time-series data into mean values per year (for each of 2020, 2021, and 2022) in order to enable exploration with the annual coverage dataset.

After excluding line items with one or more missing values, the final dataset for exploration was composed of twenty-eight explanatory variables and 154 countries with 4 years of data per country (i.e., 616 line items).

We hypothesised that different combinations of variables may help explain coverage declines in different contexts, e.g., small countries with large COVID-19 health impact and less health financing available may have sustained coverage declines. Discriminant analysis would be the designed approach for identifying linear combinations of predictors best predicting country performance classification [30,31] in the absence of correlation between the predictors. The presence of strong correlations between several predictors (Appendix A) led us to use Discriminant Analysis of Principal Components (DAPC) [30] instead, as it was designed to address this very issue. Briefly, in DAPC, a principal component analysis (PCA [32,33,34]) is first performed to preprocess (and orthogonalise) input predictors, followed by a discriminant analysis on the resulting principal components. Here, the PCA step was achieved on centred and scaled data, to account for the different scales of the predictors. Repeated cross-validation with 100 replicates and a training set including 70% of randomly selected data points were used to assess the validity of group prediction by the DAPC and to choose the optimal number of retained components in the PCA step. We report the DAPC model fit in terms of accuracy of classification per category on the test dataset and the findings of the variables identified as key contributors.

To explore the existence of nonlinear relationships between predictors and RI performance classification, we conducted a random forest analysis [35] on the same dataset (same 70:30 train/test split). We tuned the parameters of the model by exploring a range of tree sizes (0–1000) and numbers of splits (2–10) at each node and selected the parameter with the highest accuracy—which was a forest of 150 trees with ten splits at each node. We report the random forest model fit in terms of accuracy of classification per category on the test dataset, and the findings of the variables identified through importance measurements: (i) the mean decrease in accuracy (MDA) when the predictor was excluded and (ii) the mean decrease in the Gini index which assesses the purity of nodes and splits within the trees (mean decrease in impurity, MDI).

We share code and all datasets on GitHub for full reproducibility [36].

## 3. Results

### 3.1. Coverage Trends

Using *t*-tests, evidence was found of global DTP3 coverage declines vs. expectations in all years, with greatest decline in 2021—see Table 2 and Figure 1C,D. However, when translated into number of immunisations, a *t*-test on these trends over time showed compelling evidence of declines in number of immunisations primarily in 2021, then some evidence of rebounds to expected levels across the global target population of surviving infants in 2023—see Table 3.

Visual comparison in Figure 1B,D indicated the contribution of global demographic trends: the total target population of surviving infants (red line) has reduced by around ten million children since 2015. Comparison of mean coverage and number of immunisations indicated the additional contribution of relative population size, i.e., some small states must be experiencing continued coverage reductions.

DTP1 results in Figure 1A,B were directionally the same and even stronger than DTP3: point estimates of global coverage deltas were smaller, and for both 2022 and 2023, the delta in number of missed immunisations was within expected ranges. Tables of DTP1 coverage and missed immunisation *t*-test results per year are available in Appendix A.

At the country level, there was evidence that thirty-five countries out of the 190 modelled had coverage different than expected in 2023 for DTP3, and only three of these thirty-five were above expectations (Haiti, Brazil, and Mauritania). Plotting 2023 country classification on a map indicated wide geographic dispersion of countries that have not yet recovered to pre-COVID coverage trends (Figure 2).

See Appendix A for a list of countries with coverage below expectations in 2023, ranked by delta in number of immunisations.

### 3.2. Predicting RI Performance

DAPC cross-validation indicated that the optimal number of PCA axes to be retained was the maximum (Appendix A) and that this provided better-than-random predictions of RI classification 82% of the time. The predictive power was limited: the model tended towards overclassification of most countries as within expected ranges and particularly struggled to classify countries as below expectations—98.2% class error, see Appendix A for confusion matrix). Of countries performing above expectations, 55% were classified correctly. Notably, the inclusion of the nine data points where countries reported higher coverage—and where model fitting may have been poor and unrepresentative (e.g., Haiti and Brazil)—may have skewed the findings, since the set of explanatory variables differed completely when including these nine countries or not. When classifying into three categories, the contributing explanatory variables were prepandemic health system and immunisation system strength. When removing the above-expectations category and classifying countries into two categories, the key explanatory factors were government health financing, public transit closure, and mask wearing mandates. The scatterplot of the DAPC (Figure 3) confirmed that there was a considerable overlap between countries in different categories—particularly those with lower-than-expected coverage (magenta) or within expectations (gold)—and that only a few data points could be safely predicted.

When classifying into three categories, the variables identified as having explanatory power were health system strength and prepandemic immunisation system performance. When excluding the countries that performed higher than expected, the same classification accuracy challenges were found, but the explanatory predictors were government health financing, public transit closure stringency, and mask wearing mandate stringency. This lack of consistency suggests the DAPC was over-indexing to a few datapoints that swung the results.

Similarly, the random forest analysis classified most countries as within expectations and struggled to classify countries as below or above expectations (confusion matrices for the train and test datasets in Appendix A). The random forest did not identify any standout explanatory variables, with 11/28 variables having a relatively comparable mean decrease in accuracy (%) approximately or over 5% (Appendix A). These variables included population size, multiple financial indicators (GDP, all three components of health financing—government, external, and private), health system descriptors (number of nurses, health system strength, global health security, prepandemic immunisation system strength), pandemic policies (stay-at-home orders, elderly sheltering), and excess mortality.

Together, the low classification accuracy for identifying countries with coverage above or below expectations indicates that the country-level indicators considered here are unable to explain the discrepancies between expected and observed coverage performances.

## 4. Discussion

We report evidence that RI coverage is trending towards prepandemic expected levels, with stronger evidence for DTP1 than DTP3. The rebound is clearer when considering number of missed immunisations, but it is worth highlighting that there remains robust evidence that around 20% (32/190) of countries had coverage below expectations in 2023, even now that pandemic restrictions have been lifted. There are thus growing numbers of missed children to catch up from during, and now after, the pandemic in some countries. The Big Catch-Up [37], launched by WHO, UNICEF, and Gavi aims to catch-up, restore, and strengthen immunisation programmes. However, as Figure 2 helps visualise, of the thirty-two countries below confidence intervals expected coverage in 2023, only eight (Benin, India, Democratic People’s Republic of Korea, Kyrgyzstan, Mozambique, Senegal, Sudan, and Uganda) remain Gavi-eligible (excluding some additional support for middle income countries). Further (potentially self-funded) interventions are thus required in non-Gavi countries too (e.g., PAHO countries and high-income countries). Reporting on the number of missed children reached through the Big Catch-Up, and other efforts will be essential to understanding the residual immunity gap from the pandemic.

Some directional insights are possible from our descriptive analyses. Comparing trends between mean percentage coverage and number of immunisations over time highlighted the influence of declining fertility on target infant populations for immunisation. Similarly, comparing trends between expected and reported coverage for DTP1 and DTP3 was also encouraging with regards to efforts to reduce numbers of zero dose children. However, both examples highlight the need for detailed subnational geospatial analyses (e.g., in Nigeria [38]) and more granular time-series analyses in order to understand the potentially heterogeneous impact that may be obscured through these nationwide, annual datasets. Such research could be prioritised in countries with aggregate coverage declines to help provide operational insights in missed populations and guide vaccine programmes implementation.

We were not able to identify descriptive factors with compelling evidence of linear or nonlinear association with RI coverage performance through DAPC or random forest analyses. Our ecological analyses at a population level may be insufficiently detailed to understand more complex variation within countries. It may be that wide confidence intervals from the ARIMA modelling and/or simplification of complex parameters into annual population averages obscured some associations. Perhaps none of the factors explored may provide strong explanatory power—this would be interesting, e.g., it could suggest that the number of immunisations offered in an infant RI schedule is not associated with RI disruptions, indicating that a programme’s breadth does not place additional demands. Alternatively, other predictors may be required to understand drivers of continued coverage declines postpandemic. One potential such driver could be vaccine hesitancy, based on studies of attitudes towards COVID-19 vaccinations [39,40] and childhood immunisations influenced by the pandemic [41,42]. We explored including vaccine hesitancy as an explanatory variable, using a proxy based on COVID-19 vaccine mandate stringency; however, the dataset (and alternatives) was insufficiently complete.

Overall, these findings primarily indicate gaps for future research to address. Building resilient health systems that can buffer epidemic and pandemic disruptions will be essential to maintaining progress to global immunisation objectives and minimising the risks of outbreaks from vaccine-preventable diseases. Monitoring and reporting on more specific variables (e.g., health facility opening status), and subnational data could help identify drivers of coverage declines. The fact that coverage has not rebounded in all countries postpandemic also raises the question as to what is driving continued coverage declines and stagnation, since factors reported as driving RI disruption (e.g., public transport restrictions, fear of catching COVID-19, COVID-19 vaccination campaigns [12,13]) did not occur in 2023 and yet coverage did not revert to forecast levels. We encourage investigation into the determinants of continued vaccine coverage declines postpandemic.

## Figures and Tables

**Figure 1 vaccines-13-00388-f001:**
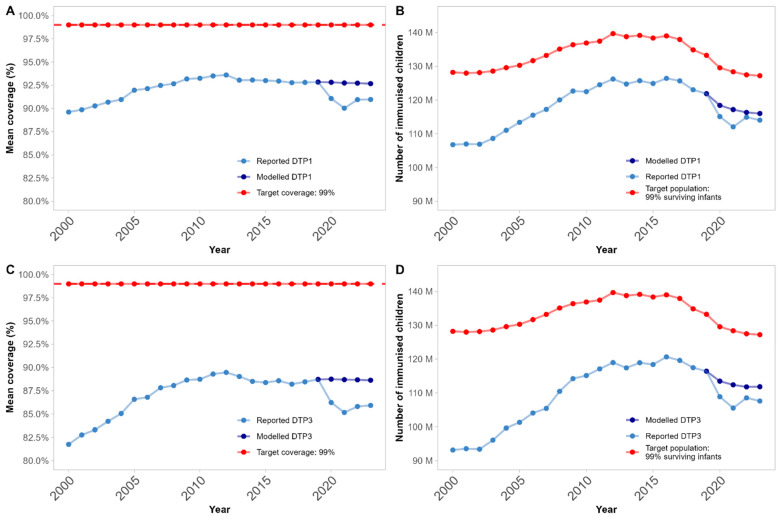
DTP1 (**A**,**B**), and DTP3 (**C**,**D**) trends overtime: (**A**,**C**) show mean global coverage, and (**B**,**D**) show total numbers of immunised children, per year from 2000 to 2023. Light blue indicates WUENIC-reported coverage in (**A**) and derived number of immunisations based on population data in (**B**); dark blue indicates ARIMA-modelled expected coverage from 2020–2023 and associated number of immunisations based on population figures; and red indicates a proxy target coverage of 99% vaccination and associated number of children.

**Figure 2 vaccines-13-00388-f002:**
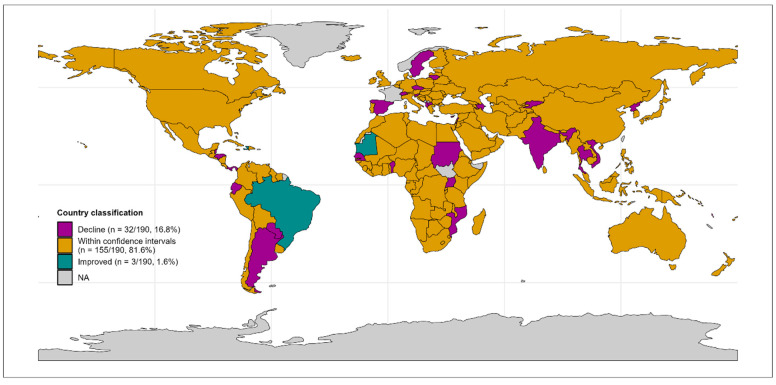
Global map illustrating DTP3 coverage performance compared with expectations in 2023 per country: light grey countries lacked sufficient data for inclusion; gold countries were within expected ranges (155 countries, 81.6%); cyan countries improved above expectations from prepandemic trends (3 countries, 1.6%); and magenta countries were below expectations (32 countries, 16.8%). All classification was based on comparison of WUENIC reported data to 95% confidence intervals from the ARIMA-modelling.

**Figure 3 vaccines-13-00388-f003:**
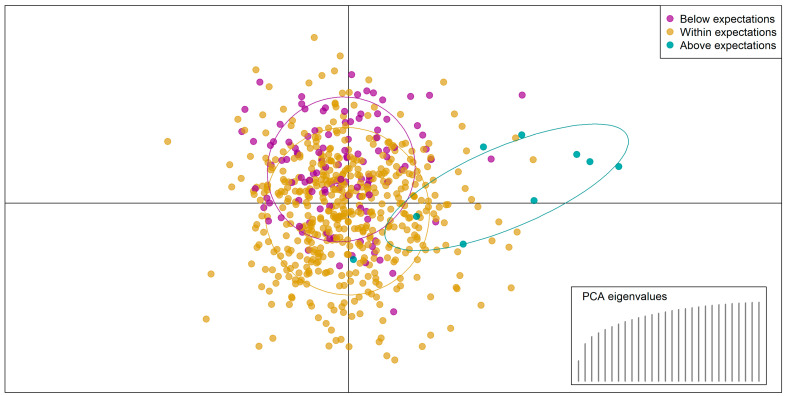
Scatterplot of discriminant analysis of principal components: the *x*-axis and *y*-axis show the first two principal components of predictors. Magenta dots indicate countries where coverage was below expectations; gold dots indicate countries where coverage was within expected ranges; and cyan dots indicate countries that were above expectations. The overlap between the groups indicates the lack of differentiation between classes in underlying predictor variables, as clusters cannot be easily distinguished.

**Table 1 vaccines-13-00388-t001:** Predictors explored for association with country coverage classification. ‘*n*’ refers to the number of predictors in a given category.

Field	Variables Included	Source
Health system descriptors (*n* = 6)	Prepandemic immunisation system strength: mean DTP3 coverage (mean 2015–2019)	WUENIC [19]
	Immunisation system breadth: number of vaccines in infant immunisation schedule (latest data)	WHO [20]
	Broader health system strength: Universal Health Coverage index (2019)	WHO [21]
	Health workforce capacity: including (i) mean number of doctors and (ii) mean number of nurses (mean 2015–2019 per 100,000 population)	WHO [22]
	Global health security index (2019)	GHS Index [23]
Financial indicators (*n* = 4)	Health financing: broken into (i) government expenditure, (ii) external (donor) investment, and (iii) private financing, i.e., out-of-pocket payments and certain insurance (mean 2015–2019 USD purchasing power parity, PPP)	WHO [24]
	Country wealth: gross domestic product (GDP; mean 2015–2019 per capita)	World Bank [25]
Pandemic impact (*n* = 17)	COVID-19 direct health burden: proxy based on number of excess deaths per 100,000 people per year (2020–2022)	*Economist* [26,27]
	Eight containment policies per year (2020–2022): stringency of (i) school closures, (ii) workplace closures, (iii) cancellation of public events, (iv) restrictions on gatherings, (v) public transport closures, (vi) stay-at-home orders, (vii) internal movement restrictions, (viii) international travel controls	Oxford COVID-19 Government Response Tracker [28,29]
	Two economic policies per year (2020–2022): extent of (i) income support and (ii) debt relief during the pandemic	Oxford COVID-19 Government Response Tracker [28,29]
	Six health policies per year (2020–2022): extent of (i) public information campaigns, (ii) COVID-19 lab/diagnostic testing policies, (iii) contact tracing efforts, (iv) mask wearing requirements, (v) availability of COVID-19 vaccines, and (vi) protection of elderly populations	Oxford COVID-19 Government Response Tracker [28,29]
Country descriptors (*n* = 1)	Population: total population (mean 2020–2023)	UNWPP [11]

**Table 2 vaccines-13-00388-t002:** Global mean expected (ARIMA-modelled) and reported DTP3 coverage from 2020 to 2023, and the calculated difference between the two (delta = expected—reported); 95% confidence intervals (CIs) and associated *p*-values from *t*-tests.

Year	Expected	Reported	Delta [95% CIs]	*p*-Value
2020	88.7%	86.2%	−2.5% [−1.7%; −3.3%]	<0.0001
2021	88.7%	85.2%	−3.5% [−2.3%; −4.7%]	<0.0001
2022	88.7%	85.8%	−2.9% [−1.4%; −4.3%]	0.0002
2023	88.6%	85.9%	−2.7% [−1.1%; −4.3%]	0.0008

**Table 3 vaccines-13-00388-t003:** Number of expected (ARIMA-modelled) and reported (from WUENIC) DTP3 immunisations per year globally from 2020 to 2023, and the calculated difference between the two (delta = expected—reported); 95% confidence intervals (CIs) and associated *p*-values reported from *t*-test.

Year	Expected	Reported	Delta [95% CIs]	*p*-Value
2020	597,036	572,857	−24,180 [−4755; −43,605]	0.02
2021	591,324	555,302	−36,022 [−10,597; −61,448]	0.006
2022	588,180	571,111	−17,069 [−3366; −30,772]	0.02
2023	588,432	566,191	−22,240 [−212; −44,269]	0.05

## Data Availability

All data and code for full reproducibility are available on GitHub: https://github.com/bevans249/ri_post_pandemic (accessed on 21 March 2025).

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
