# Peer review of "Routine Immunisation Coverage Shows Signs of Recovery at Global Level Postpandemic, but Important Declines Persist in About 20% of Countries"

_vaccines, 2025, doi:10.3390/vaccines13040388_

Round 1
Reviewer 1 Report
Comments and Suggestions for Authors
This is a very interesting analysis of post-pandemic immunization coverage rates, recovery trends and potential predictors for the degree of on-going disruption with a sophisticated, advanced statistical methodology. Although I am not a statistician and cannot judge the validity of the statistical results completely, my impression is that the results are robust and add new insights to the scientific data base.
Following below are my mostly minor comments.
Title: shouldn't the second half of the title read "but important declines persist in 20% of countries" without "and"?
Abstract: it summarizes the full article well. In line 31 the authors write "17.6%" of countries have not reached yet expected levels, but in line 35 they write "20%". Perhaps they should insert "close to" or "about" before "20%"?
Introduction: in line 56-57 I suggest to add the remark that immunization coverage is measured using DTP3 as the main proxy indicator (in line with WHO guidance).
In line 69, the sentence reads "as drivers of interruptions as drivers of disruption". I suggest to stick to one term, interruption or disruption.
In line 77-78 the expression "compared to during (2020-2022) and pre-pandemic (2000-2019)" sounds somehow odd for me as a non-native English speaker.
In line 80 DTP3 is mentioned the first time in the main text; please provide the full term (only presented in the abstract so far).
Materials and Methods: in line 88 "ARIMA" is mentioned as a statistical method for the first time. Please mention that this means "autoregressive integrated moving average" and include it in the list of abbreviations.
Lines 119-120 and table 1: I found this statistical approach a bit difficult. That the health system descriptors were all taken from the pre-pandemic period and that the authors constructed a mean is understandable. But to present the pandemic impact descriptors as a mean value of the years 2020-2022 is not so clear. During those years, many policies with regard to these descriptors were changed frequently, sometimes within weeks. How is it possible to calculate mean values then? This would need some more explanation. In addition, all 17 descriptors were taken from two sources, The Economist (ref. 28,29) and the Oxford COVID-19 Government Response Tracker (ref. 30). While I am questioning somehow the validity of The Economist as a source for scientific data (see also my comment on the references), I wonder why for the latter reference a publication from year 2021, well before the end of all the pandemic policies, was chosen. I suggest to add at least a link to the project website as an additional reference because the original reference 30 does not provide any data for the years 2021 and 2022.
Results: This section is well presented and describes the results in sufficient detail. I wonder whether the fact that no descriptive factors with sound evidence could be identified was due to my observation that at least the construction of the pandemic impact descriptors seems difficult. The authors could elaborate on this more in the discussion.
Discussion: The authors analyse their results in depth and offer possible reasons why they could not identify robust descriptive factors (see also my comment under Results). As they mention, it was not possible to include vaccine hesitancy as a descriptor due to missing data sets. From my experience as a practising clinician, this factor is one of the most pertinent drivers of incomplete recovery.
Abbreviations: I found at least nine more abbreviations ARIMA, WHO, UNICEF, UNWPP, GHS, GDP, MDA, MDI, CI) not listed in this section. I wonder whether a list of abbreviations is necessary at all.
References: check ref. 14: it seems to be incorrect. Do the authors mean the article published in Frontiers in Pediatrics in 2022? The title and journal details they provide are the same as in ref. 13.
Check ref. 25: provide the source (URL or something similar). Ref. 28: how valid and reliable is that analysis?
Ref. 32-24: they are rather old. Are there no more recent ones dealing with the same topics?
Ref. 39 and 40: both articles use data from the period before or of the early phase of the pandemic. Would it not be better to add articles using data from the later period of the pandemic? Possible examples: Talbert N, Wong N. In Whom We Trust: The Effect of Trust, Subjective Norms, and Socioeconomic Status on Attitudes and COVID-19 Vaccination Intentions. Health Commun. 2025 Feb 4:1-14. doi: 10.1080/10410236.2025.2456995. Matthijssen MAM, Cloin M, van Leeuwen F, van de Goor I, Achterberg P. What made people (more) positive toward the COVID-19 vaccine? Exploring positive and negative deviance perspectives. BMC Public Health. 2025 Feb 4;25(1):441. doi: 10.1186/s12889-024-21027-1.
Supplementary materials: Table S2: I think after S2: it should read "DTP3" ...
Table S3: I wonder whether the ranking by delta in number of immunizations is the correct way. Countries with large populations but small percentages in delta would still rank highest as shown for e.g. India: delta -6.9%, but delta of total number of immunisations -1.6 million. I would suggest changing the ranking to the delta as a percentage.
Author Response
Thank you for your thorough review. Please see below and attached for a point-by-point response. We hope we have fully addressed your recommendations and believe that the paper is stronger for these edits.
Comments 1: Title: shouldn't the second half of the title read "but important declines persist in 20% of countries" without "and"?
Response 1: Yes, thank you, the reviewer is correct, and this was an oversight when copying over to the Vaccines template. This is corrected and the title now reads: “Routine immunisation coverage shows signs of recovery at global-level post pandemic, but important declines persist in about 20% of countries”.
Comments 2: Abstract: it summarizes the full article well. In line 31 the authors write "17.6%" of countries have not reached yet expected levels, but in line 35 they write "20%". Perhaps they should insert "close to" or "about" before "20%"?
Response 2: Agreed, thank you. We have changed the title and line 35 of the Abstract Conclusions section to say “about 20%” as proposed.
Comments 3: Introduction: in line 56-57 I suggest to add the remark that immunization coverage is measured using DTP3 as the main proxy indicator (in line with WHO guidance).
Response 3: We have added this remark for supporting use of DTP3 as the main proxy indicator in at line 96-97 in of Section 2. Materials and Methods, which now notes we use DTP3 “to understand broad immunisation system performance trends, in line with WHO guidance”. We have made this edit in this later section, since we have removed the specificity to DTP in the introduction to optimise readability and simplicity.
Comments 4: In line 69, the sentence reads "as drivers of interruptions as drivers of disruption". I suggest to stick to one term, interruption or disruption.
Response 4: We agree and have opted for “drivers of disruption”.
Comments 5: In line 77-78 the expression "compared to during (2020-2022) and pre-pandemic (2000-2019)" sounds somehow odd for me as a non-native English speaker.
Response 5: We have simplified lines 76-78 to say our first aim is to “Quantify global and country-level immunisation coverage trends in the pandemic and post-pandemic period (years: 2020-2023) compared to pre-pandemic (years: 2000-2019), in terms of percentage coverage and number of immunisations”.
Comments 6: In line 80 DTP3 is mentioned the first time in the main text; please provide the full term (only presented in the abstract so far).
Response 6: We have replaced the specificity of DTP to “routine immunisation” in this section (now line 79). The full term is provided at line 93 for what is now the first usage of DTP.
Comments 7: Materials and Methods: in line 88 "ARIMA" is mentioned as a statistical method for the first time. Please mention that this means "autoregressive integrated moving average" and include it in the list of abbreviations.
Response 7: Both of these recommendations are actioned – line 88-89 (re-numbered due to previous edits) details the full term, and we have included ARIMA in the Abbreviations table at end.
Comments 8: Lines 119-120 and table 1: I found this statistical approach a bit difficult. That the health system descriptors were all taken from the pre-pandemic period and that the authors constructed a mean is understandable. But to present the pandemic impact descriptors as a mean value of the years 2020-2022 is not so clear. During those years, many policies with regard to these descriptors were changed frequently, sometimes within weeks. How is it possible to calculate mean values then? This would need some more explanation.
Response 8: We clarify that we have constructed yearly averages for each predictor with data covering the pandemic period (i.e., three values, one for mean in 2020, another for mean in 2021, and a third for mean in 2022). We have clarified in lines 122-125 to read “For indicators reporting variables during the pandemic period, we synthesised more granular time series data into mean values per year (for each of 2020, 2021, and 2022) in order to enable exploration with the annual coverage dataset.”.We are necessitated to calculate annual summary terms due to the limitation that our response variable (DTP3 coverage) is only available from WUENIC as an annual estimate. Whilst policies did vary significantly over time, we do note – hopefully reassuringly – that variation is still seen between countries in the stringency of policies when summarised at an annual level.
Comments 9: In addition, all 17 descriptors were taken from two sources, The Economist (ref. 28,29) and the Oxford COVID-19 Government Response Tracker (ref. 30). While I am questioning somehow the validity of The Economist as a source for scientific data (see also my comment on the references).
Response 9: We explored use of the four available and often-cited excess mortality datasets to use as a proxy for direct COVID-19 health impact (given the challenges with case and death reporting being inconsistent across countries). The Economist dataset was the most complete in terms of covering all countries, and each year from 2020 to 2022 – allowing us to use a consistent source for this variable enabling between country comparisons. The other estimates (WHO, World Mortality Dataset, and COVID-19 Excess Mortality Collaborators) were either limited to a subset of countries (127 countries for World Mortality Dataset) and/ or only available for 2020-2021 (WHO, World Mortality Dataset) and/ or concerns raised with lack of alignment with other methods and estimates (COVID-19 Excess Mortality Collaborators – https://www.thelancet.com/journals/lancet/article/PIIS0140-6736(23)00112-5/fulltext).
A recent paper by De Nicola et al. 2025 (https://academic.oup.com/jrsssa/article/188/1/205/7639023) has compared these excess mortality estimates in high-income countries in the pandemic (note: they refer to COVID-19 Excess Mortality collaborators as IHME). There is some variation, however the Economist estimates appear internally consistent (i.e., consistently lower than some estimates and higher than others) and directionally in-line with other (non-IHME) methods.
Comments 10: I wonder why for the latter reference a publication from year 2021, well before the end of all the pandemic policies, was chosen. I suggest to add at least a link to the project website as an additional reference because the original reference 30 does not provide any data for the years 2021 and 2022.
Response 10: To aid reader reproducibility have now also added a reference directly to the Oxford COVID-19 Government Response Tracker GitHub dataset (https://github.com/OxCGRT/covid-policy-dataset). This is inserted in Table 1 in the relevant rows. We also note that the OxCGRT recommend the use of the 2021 publication as the reference to cite when using the data in academic papers, so we have left this in too.
Comments 11: Results: This section is well presented and describes the results in sufficient detail. I wonder whether the fact that no descriptive factors with sound evidence could be identified was due to my observation that at least the construction of the pandemic impact descriptors seems difficult. The authors could elaborate on this more in the discussion.
Response 11: We are very much agreed that more detailed time series analyses, dependent on access to more granular coverage datasets, could be very insightful. To make this clearer we have added “the need for… more granular time series analyses in order to understand the potentially heterogeneous impact that may be obscured through these nationwide, annual datasets” in lines 289-291 in section 4. Discussion. Additionally we have reframed for clarity another sentence in the Discussion to highlight these issues (lines 299-301): “It may be that wide confidence intervals from the ARIMA modelling and/ or simplification of complex parameters into annual population averages obscure some associations.”. Researchers or programme managers in countries, with access to monthly administration data (for example) could use similar methods to ours to explore seasonal and pandemic trends in coverage that we are not able to (when relying on publicly available validated annual data).
Comments 12: Abbreviations: I found at least nine more abbreviations ARIMA, WHO, UNICEF, UNWPP, GHS, GDP, MDA, MDI, CI) not listed in this section. I wonder whether a list of abbreviations is necessary at all.
Response 12: For completeness, we have included all of these abbreviations to the Abbreviations table.
Comments 13: References: check ref. 14: it seems to be incorrect. Do the authors mean the article published in Frontiers in Pediatrics in 2022? The title and journal details they provide are the same as in ref. 13.
Response 13: Thank you for this thorough checking. In the reference manager the double-barreled surname of the first author “Cardoso Pinto” had somehow resulted in two almost duplicate references (one for Cardoso, and one for Pinto – both referencing the same paper, as the author correctly noted!). This is now corrected to remove the duplication (and correctly list the first author as Cardoso Pinto).
Comments 14: Check ref. 25: provide the source (URL or something similar).
Response 14: Upon checking, this reference was directed towards the same website as the previous reference, i.e., the GHS Index https://ghsindex.org/. We have therefore kept only the first reference, including the URL.
Comments 15: Ref. 28: how valid and reliable is that analysis?
Response 15: Copying part of our response above for ease, since relevant here. A recent paper by De Nicola et al. 2025 (https://academic.oup.com/jrsssa/article/188/1/205/7639023) has compared these excess mortality estimates in high-income countries in the pandemic (note: they refer to COVID-19 Excess Mortality collaborators as IHME). There is some variation, however the Economist estimates appear internally consistent (i.e., consistently less than some estimates and more than others) and directionally in-line with other (non-IHME) methods.
Comments 16: Ref. 32-24: they are rather old. Are there no more recent ones dealing with the same topics?
Response 16: These papers are the original references for the Principal Component Analysis and the Discriminant Analysis. While there are indeed hundreds of more recent references, it is customary to cite the original paper introducing a statistical method. We have now also added a reference to a more recent textbook on multivariate analysis (Legendre & Legendre, 2012).
Comments 17: Ref. 39 and 40: both articles use data from the period before or of the early phase of the pandemic. Would it not be better to add articles using data from the later period of the pandemic? Possible examples: Talbert N, Wong N. In Whom We Trust: The Effect of Trust, Subjective Norms, and Socioeconomic Status on Attitudes and COVID-19 Vaccination Intentions. Health Commun. 2025 Feb 4:1-14. doi: 10.1080/10410236.2025.2456995. Matthijssen MAM, Cloin M, van Leeuwen F, van de Goor I, Achterberg P. What made people (more) positive toward the COVID-19 vaccine? Exploring positive and negative deviance perspectives. BMC Public Health. 2025 Feb 4;25(1):441. doi: 10.1186/s12889-024-21027-1.
Response 17: Thank you for directing us to these more recent papers. This was also a good prompt for us to search for more recent publications on the effect of the pandemic on attitudes towards childhood vaccinations (since this may not correlate with their attitude towards COVID-19 vaccination for themselves!). We have included the reviewer’s second proposed citation as well as an additional study (Skirrow et al., 2024) that explored the impact of COVID-19 on UK parent’s attitudes towards routine immunisation. Lines 305-307 now read: “One potential such driver could be vaccine hesitancy – based on studies of attitudes towards COVID-19 vaccinations [38,39] and childhood immunisations influenced by the pandemic [40,41] …”
Comments 18: Supplementary materials: Table S2: I think after S2: it should read "DTP3" ...
Response 18: This table is showing the number of immunisations for DTP1, which was explored as a secondary vaccine touchpoint to look at impact on Zero Dose children. The equivalent for DTP3 is reported in the main body of the paper as Table 3. Therefore, we have made no changes.
Comments 19: Table S3: I wonder whether the ranking by delta in number of immunizations is the correct way. Countries with large populations but small percentages in delta would still rank highest as shown for e.g. India: delta -6.9%, but delta of total number of immunisations -1.6 million. I would suggest changing the ranking to the delta as a percentage.
Response 19: We were also torn by this consideration. Ultimately, we decided that if considering where more efforts are needed to reach the most affected children, then it was useful to see in terms of absolute numbers. Otherwise, we would see small states like Vanuatu as the most affected – yet these small states can often have larger swings in coverage levels over time as seen in ARIMA time series model fitting. We propose leaving as is – but do appreciate the consideration given on how to present this information and promote action.

Reviewer 2 Report
Comments and Suggestions for Authors
This is a hugely interesting study and will be important to understand why countries perform the way they do.
But - didnt understand the objective. Title itself has a word missing - important ....?
Also, is this only part of a study? Why were so many countries left out? How did you select them?
A lot of work needs to be added to this report - and it is worth it
Comments on the Quality of English LanguageNeeds work for proper sentence construction.
No intro.
Title itself grammatically incorrect
Author Response
Comments 1: Title itself has a word missing - important ....?
Response 1: Apologies, there was an oversight when copying over the title to the Vaccines template. This is corrected and the ‘and’ is removed to give the correct title: “Routine immunisation coverage shows signs of recovery at global-level post pandemic, but important declines persist in about 20% of countries”.
Comments 2: Also, is this only part of a study? Why were so many countries left out? How did you select them?
Response 2:
Our analyses are conducted on all countries/states globally with complete data for each stage of the modelling. There are three stages to this ecological analysis, as described in the section 2 Methodology.
The first is a coverage trends analysis, which is conducted on all countries/states included in the WHO and UNICEF Estimates of National Immunisation Coverage dataset with complete data from 2000-2019 – namely 190 countries. Whilst this was previously detailed in the abstract (line 21: “…time-series forecasting across 190 countries.”) we have added additional details to the section 2.1 Materials and Methods – Coverage trends to specify the number of countries included and rationale. Lines 89-91 now reads “…project 2020-2023 coverage levels per country for all 190 countries in the WUENIC dataset with complete time series data from 2000-2019 inclusive.”
The second stage of this analysis synthesises the coverage trends to assess aggregate global coverage trends versus expectations. We have added in details in section 2.2 Materials and Methods – Global and country-level performance lines 100-101 to confirm that these are the same set of countries as for the coverage trends forecasting: “To assess global coverage trends, we conducted (1) t-tests on coverage deltas per year across the same 190 countries as in Section 2.1…”.
The third stage of this analysis is predictive analysis using Discriminant Analysis of principal Components (DAPC) and Random Forests to classify country immunisation performance based on 28 hypothesised explanatory variables. As described in the methods section lines 130-132 these classification analyses are conducted on a subset of 154 countries, since we have removed countries with one or missing explanatory variables: “After excluding line items with one or more missing values, the final dataset for exploration is composed of 28 explanatory variables and 154 countries with 4 years of data per country (i.e., 616 line items).”.
Comments 3: A lot of work needs to be added to this report - and it is worth it
Response 3: We have done our best to address all specific comments by all referees. We believe the revision has gained in clarity and details.
Comments 4: Needs work for proper sentence construction.
Response 4: The lead author is a native English speaker. We have fixed the typo in the title, and proof-read the manuscript entirely. We double-checked using Word spell & grammar check which gives an Editor score of 99%.
Comments 5: No intro.
Response 5: The introduction is found in section 1, lines 40-82.
Comments 6: Title itself grammatically incorrect
Response 6: As described above, this was an oversight from copying over the draft manuscript to the Vaccines template. This is now corrected in the revision and the correct title reads: “Routine immunisation coverage shows signs of recovery at global-level post pandemic, but important declines persist in about 20% of countries”.

Reviewer 3 Report
Comments and Suggestions for Authors
This is an excellent and much needed study. The authors went to a great deal of effort to identify appropriate prediction databases and carefully chose and explained appropriate statistical analyses.
Their conclusions are important and well-supported by these statements:
“Together, the low classification accuracy for identifying countries with coverage above or below expectations indicates that the country-level indicators considered here are unable to explain the discrepancies between expected and observed coverage performances”
“there remains strong evidence that around 20% (32/190) of countries have coverage below expectations in 2023, even now that pandemic restrictions have been lifted. There are thus growing numbers of missed children to catch-up from during and now post-pandemic in some countries. The Big Catch-Up [37], launched by WHO, UNICEF, and Gavi to close COVID-19 RI gaps, 9 of 13 aims to do this in Gavi-eligible countries. However, as Figure 2 helps visualise, of the 32 270 countries below confidence intervals expected coverage in 2023, only 9 (Kyrgyzstan, Haiti, Benin, Senegal, Democratic People’s Republic of Korea, Mozambique, Uganda, Sudan, and India) remain Gavi-eligible. Further (potentially self-funded) interventions are thus required in non-Gavi countries too (e.g., PAHO countries and high-income countries). Reporting on the number of missed children reached through the Big Catch-Up, and other efforts, will be essential to understanding the residual immunity gap from the pandemic.”
“highlights the influence of declining fertility on target infant populations for immunisation. Similarly, comparing trends between expected and reported coverage for DTP1 and DTP3 is also encouraging with regards to efforts to reduce numbers of Zero Dose children. However, both examples highlight the need for detailed sub-national geospatial analyses (e.g., in Nigeria [38]) in order to understand the potentially heterogeneous impact that may be obscured through these nationwide, annual datasets.”
Their comment about variations within countries due to areas of higher poverty are important and expected.
Author Response
Thank you for the positive words in support of the approaches used in these analyses and the value of the outputs. We appreciate your feedback. We have done our best to address all specific comments by all referees. We believe the revision has gained in clarity and details.